# Lack of Association between Cytokine Genetic Polymorphisms in Takayasu’s Arteritis in Mexican Patients

**DOI:** 10.3390/ijerph16234863

**Published:** 2019-12-03

**Authors:** María Elena Soto, Claudia Huesca-Gómez, Yazmín Torres-Paz, Giovanny Fuentevilla-Álvarez, Ricardo Gamboa

**Affiliations:** 1Immunology Department, Instituto Nacional de Cardiología “Ignacio Chávez”. Juan Badiano No. 1, Col. Sección XVI, Tlalpan, México City 14080, Mexico; sotele@cardiologia.org.mx; 2Physiology Department, Instituto Nacional de Cardiología “Ignacio Chávez”, Juan Badiano No. 1, Col. Sección XVI, Tlalpan, Mexico City 14080, Mexico; claudia.huesca@cardiologia.org.mx (C.H.-G.); yazminestela@hotmail.com (Y.T.-P.); fuentevilla_alvarez@hotmail.com (G.F.-Á.)

**Keywords:** cytokines, Takayasu′s arteritis, polymorphisms, Mexican population

## Abstract

*Aim*: To investigate the relation between polymorphisms in the interleukin 10 (*IL*)-*10*, tumor necrosis factor (*TNF)*-α, transforming growth factor (*TGF)-*β and interferon (*IFN)-γ* genes and Takayasu’s arteritis in the Mexican population. *Methods*: A case-control study was performed to investigate the associations of *IL-10*, *TNF-α*, *TGF-β* and *IFN-γ* polymorphisms in a sample of 52 Takayasu’s arteritis patients, diagnosed according to the criteria of the American College of Rheumatology and EULAR PRINTO criteria when the patients were under 18 years of age; 60 clinically healthy unrelated Mexican individuals by the 5′ exonuclease TaqMan polymerase chain reaction. Polymorphic haplotypes were constructed after linkage disequilibrium analysis. *Results:* Significant differences were not found in the distribution for genotype and allele frequencies of the polymorphisms studied between healthy controls and Takayasu´s arteritis patients. Likewise, significant associations were not detected in the haplotype analysis with the different genes studied. *Conclusions*: These findings suggest that the polymorphisms in *IL-10*, *TNF-α*, *TGF-β* and *IFN-γ* might not contribute to the susceptibility of Takayasu´s arteritis in the Mexican population.

## 1. Introduction

Takayasu′s arteritis (TA) is an inflammatory disease that affects medium and large arteries, predominantly the aorta and its main branches. The arterial inflammation can lead to wall thickening, arterial stenosis, fibrosis, thrombus formation, and progressive occlusion [1]. The clinical manifestations of Takayasu´s arteritis usually appear in childbearing-age women [1,2].

Inflammation in Takayasu´s arteritis begins around the vasa vasorum and it is accompanied by the infiltration of several inflammatory cells, leading to granuloma formation. At this stage, the production of inflammatory mediators is markedly increased [3,4]. This response includes many factors such as CD4^+^/CD8^+^ lymphocytes, macrophages and pro-inflammatory cytokines.

Cytokines are involved in synergistic and antagonistic interactions, exhibiting both positive and negative regulatory effects [5]. Several studies have related single nucleotide polymorphisms (SNPs) of some cytokine genes to risk factors in inflammatory processes. The formation of granulomas in giant cell arteritis and vascular granulomatous are also related [6,7]. Among the cytokines that might be involved, tumor necrosis factor (TNF)-α, a powerful immunomediator and proinflammatory cytokine that influences a wide variety of adverse effects on the body’s cells, can activate growth factors, acting as a chemoattractant and affecting the synthesis of adhesion molecules.

Several studies have suggested that the expression of TNF-α is affected by polymorphisms in the promoter region of its gene, which has been identified at position-238 (rs361525) and-308 (rs1800629). These polymorphisms modify the production of the cytokine. On the other hand, transforming growth factor (TGF)-β is a cytokine that induces the de novo differentiation of IL-17-producing T cells in the presence of pro-inflammatory cytokine IL-6 and could also be altered in TA [7]. Interleukin (IL-10) is an important anti-inflammatory cytokine, which can be secreted by TH2 cells and macrophages, having powerful deactivation properties and anti-inflammatory effects, since it inhibits the synthesis of cytokines [8]. It has been reported that approximately 75% of the difference in IL-10 secretion is due to genetic factors and controlled at the transcriptional level [9]. Three SNPs (*C-592A, C-819T, and G-1082A*) in its promoter region have been associated with several diseases such as cardiovascular diseases; however, the results are not conclusive [10,11,12,13]. Interferon-gamma (IFN-γ) is produced by Th1-type CD4+ lymphocytes, CD8+ lymphocytes, and natural killer (NK) cells, and its function is focused on macrophage inflammation intervention [14,15].

Thus, the aim of our study was to determine whether polymorphisms in the *IL-10, TNF-α, TGF*-*β* and *IFN-γ* genes are associated with the development of TA in patients, comparing these polymorphisms to those found in healthy individuals.

## 2. Materials and Methods

### 2.1. Subjects

All participants or their guardians provided a written informed consent form prior to the study. The study complied with the Declaration of Helsinki and was approved by the Ethics Committee of the Instituto Nacional de Cardiología “Ignacio Chávez” (Ethical approval number: 10-678).

The selection of patients diagnosed with Takayasu’s arteritis was carried out by the criteria of the American College of Rheumatology and then classified according to the criteria proposed by Hata et al. [16] into 5 subtypes: (1) patients with aortic arch involvement were considered as type I; (2) patients in which the lesion was limited to the ascending aorta and the aortic arch were considered as type IIa; patients in which the lesion also included the descending aorta without involvement of the celiac artery were considered as type IIb; (3) patients in which the lesion involved the descending aorta (from the end of the aortic arch to the femoral artery) were considered as type III (4) patients with damage to the abdominal aorta and renal arteries were classified as type IV; (5) patients with involvement of the entire aorta and its branches were classified as type V. In each type, coronary and pulmonary arteries may be involved. [16,17].

For children aged 18 years or younger, we used EUKAR PRINTO criteria: classification of TA required typical angiographic abnormalities of the aorta or its main branches and pulmonary arteries (mandatory criterion) plus one of five criteria—(1) pulse deficit or claudication; (2) blood pressure discrepancy in any limb; (3) bruits; (4) hypertension; (5) elevated acute phase reactant [18].

In addition, 60 clinically healthy patients were studied. These subjects were without any kinship and were recruited at the National Cardiology Institute ¨Ignacio Chávez¨ in Mexico City. The inclusion criteria for control subjects were: normal parameters of body mass index (BMI) and plasma lipid levels, absence of hypertension, familial histories of type 2 diabetes mellitus, coronary heart disease or inflammatory-associated diseases.

For all participants, blood pressure was measured after 5 min in a sitting position. The criteria for hypertension was the following; diastolic pressure above 90 mmHg and systolic pressure above 140 mmHg in at least three recordings on different days.

The National Institute of Cardiology “Ignacio Chávez” is a reference center for Takayasu’s arteritis. Consequently, the patients came from different states in Mexico. All participants were unrelated and were of Mexican Mestizo descent—i.e., the individual and the last three generations of their family were born in Mexico. A Mexican Mestizo is defined as someone born in Mexico, who is a descendant of the original native inhabitants of the region and individuals, mainly Spanish, of Caucasian and/or African origin, who arrived in America during the sixteenth century.

### 2.2. DNA Preparation

Genomic DNA was extracted from whole blood containing EDTA by standard techniques. The IL-10 −1082 A/G (rs1800896), IL-10 −819 C/T (rs1800871), IL-10 −592 A/C (rs1800872), TNF-α −238 A/G (rs361525), TNF-α −308 A/G (rs1800629), IFN-γ −179 G/T (rs2069709), IFN-γ −155 G/A (rs2069710), TGF-β −509 T/C (rs1800469) and TGF-β 29 T/C (rs1800470) SNPs were genotyped using 5′ exonuclease TaqMan genotyping assays on a 7900HT Fast real-time PCR system, according to the manufacturer’s instructions (Applied Biosystems, Foster City, CA, USA). Each SNP (allele and genotype) was manually and automatically defined with allelic discrimination software (7300 System SDS Software^®^ by Applied Biosystems, Foster City, CA, USA) (Table 1).

### 2.3. Statistical Analysis

All calculations were performed using SPSS version 18 (SPSS Chicago, Il, USA) and EPISTAT statistical program (Version 5.0; USD Incorporated 1990, Stone Mountain, Georgia).

The p values were corrected (pC) according to the number of specificities tested and the number of comparisons performed, and they were considered statistically significant if their value was <0.05. Relative risk with 95% confidence intervals (CI) was calculated as the odds ratio. Pairwise linkage disequilibrium (LD, D’) estimations between polymorphisms and haplotype reconstruction were performed with Haploview version 4.1 (Broad Institute of Massachusetts Institute of Technology and Harvard University, Cambridge, MA, USA). Statistical significance was accepted at an alpha level of less than or equal to 0.05.

## 3. Results

A total of 112 subjects were analyzed, including 52 Takayasu´s arteritis patients according to the American College of Rheumatology criteria. Thirty-two subjects were in the active phase (61.5%) and 20 were in the nonactive phase (38.5%). Hypertension was present in 24 (46.1%) TA patients (one with type I Hata’s classification, three with type II and 20 with type V). The control group included 48 women and 12 men, with a mean age of 34.7 years and an range of 17–51 years. The demographic characteristics are shown in Table 2. The observed and expected frequencies in both groups were in the Hardy–Weinberg equilibrium, indicating that our population does not have co-dominance or linkage disequilibrium.

Table 3 summarizes the allele and genotype frequencies of the *IL-10* −*1082 A/G* (rs1800896), *IL-10* −*819 C/T* (rs1800871), *IL-10* −*592 A/C* (rs1800872), *TNF-α* −*238 A/G* (rs361525), *TNF-α* −*308 A/G* (rs1800629), *IFN-γ* −*179 G/T* (rs2069709), *IFN-γ* −*155 G/A* (rs2069710), *TGF-β* −*509 T/C* (rs 1800469) and *TGF-β 29 T/C* (rs 1800470) genes in Takayasu’s arteritis and healthy control groups.

In the *TNF-α* gene in both studied polymorphisms (rs361525 and rs1800629), no homozygous individuals were found for the minor allele; the same was true for both the studied polymorphisms in *IFN-γ* (rs2069709 and rs2069710). In the case of *TGF-β* −*509 T/C* (rs1800469), an increase in the frequency of the heterozygous genotype was observed. However, there was no statistical difference. On the other hand, in the analysis of *IL-10* polymorphisms, (rs1800872) and *IL-10* −*819 C/T* (rs1800871), the most frequent genotypes were heterozygous and, in both cases, the predominant allele was the C allele. For *IL-1082* (rs1800896), the most frequent allele was the A allele. In all cases, none of the genotypes studied showed statistically significant differences between the two groups.

In addition, we analyzed other inheritance models (additive and recessive). For *TGF-β* −*509*, *p* = 0.609 in the additive model, while *p* = 0.381 in the recessive model. For *TGF-β*, there were 29 values at *p* = 0.313 and *p* = 0.691, respectively. For *IL-10* −*592*, in the additive model, *p* = 0.988, and *p* = 0.940 in the recessive model. For *IL-10* −*819*, the value was *p* = 0.730 and *p* = 774, and *p* = 0.844 and *p* = 0.336 for *IL-10* −*1082*, respectively. Finally, after the construction of the haplotypes of the genes studied, it was only possible to observe the distribution of the *IL-10, TNF-α* and *TGF-β* genes (Table 4).

The association between genetic variants in *IL-10* −*1082 A/G* (rs1800896), *IL-10* −*819 C/T* (rs1800871), *IL-10* −*592 A/C* (rs1800872), *TNF-α* −*238 A/G* (rs361525), *TNF-α* −*308 A/G* (rs1800629), *IFN-γ* −*179 G/T* (rs2069709), *IFN-γ* −*155 G/A* (rs2069710), *TGF-β* −*509 T/C* (rs 1800469) and *TGF-β 29 T/C* (rs 1800470) and TA was evaluated by investigating allele, genotype, and haplotype frequencies in patients in the active phase or non-active phase, different age onset and Hata’s classification. No significant differences were observed between TA patients in any of the aforementioned parameters.

In order to evaluate the effect of the genetic charge derived from ethnicity, a comparison was made between the allelic frequencies of our Takayasu’s patient group and other populations previously reported with healthy subjects (Table 5). A statistically significant difference was found in the three studied genotypes of *IL-10* with respect to the populations of both Korea and Tunisia, while for *TNF-α*, there were differences with the population of Taiwan and with the population of Iran in *TNF-α* −*238* as well as in *TGF-β* −*29*.

## 4. Discussion

Despite the various studies carried out, the etiological factors that lead to TA remain unknown. It has been suggested that the presence of both genetic and environmental autoimmune factors may play a role in inflammatory processes. This results in an immune response mediated by large cells. Inflammatory processes are regulated by a fine balance between the pro-inflammatory and anti-inflammatory cytokines in infectious and autoimmune diseases and have been reported in several studies [19,20,21,22]. Nevertheless, which of these components are present and how they can influence the progression of the lesions is not well known in Takayasu’s arteritis. In this study, we looked for a possible association between the polymorphisms in the *IL-10* −*1082 A/G* (rs1800896), *IL-10* −*819 C/T* (rs1800871), *IL-10* −*592 A/C* (rs1800872), *TNF-α* −*238 A/G* (rs361525), *TNF-α* −*308 A/G* (rs1800629), *IFN-γ* −*179 G/T* (rs2069709), *IFN-γ* −*155 G/A* (rs2069710), *TGF-β* −*509 T/C* (rs 1800469) and *TGF-β 29 T/C* (rs1800470) genes and the development of Takayasu’s arteritis.

Some studies have attempted to explore the association of different diseases with pro-inflammatory and anti-inflammatory cytokines. Studies have shown that pro-inflammatory cytokines, such as TNF-α, play an important role in granuloma formation and blood levels of TNF are increased in TA patients [21]. Other studies have shown that IL-10 plays a crucial role in the regulation of inflammation.

IL-10 is a potent inhibitor in the synthesis of proinflammatory cytokines; it inhibits the action of macrophages and, as a consequence, suppresses the activation of Th1 cells and adhesion molecules. It is known that IL-10 and TNF-α have complex and predominantly opposing roles in inflammation [22,23]. A self-regulating circuit has been reported in which, TNF-α stimulates the production of IL-10, which, in turn, reduces the synthesis of TNF-α [24]. This mechanism plays a fundamental role in the inhibition of immune and inflammatory responses. Many functions of IL-10 focus on the inhibition of macrophage function, including cytotoxic activity and cytokine synthesis. Thus, a low expression of IL-10 has been related to the polymorphisms in the *IL-10* gene (−1082 A/G, −819 T/C and −592 A/C) in patients with hypertension, myocardial infarction and coronary artery disease [10,25,26].TGF-β is a cytokine widely distributed throughout the body. It acts as a suppressor of cell proliferation and the migration of vascular smooth muscle cells, promoting cell differentiation and apoptosis, as well as production of the plasminogen activator inhibitor. It is thus suggested that TGF-β-1 is involved in the atherogenic process [27,28]. Reports indicate that polymorphisms (29 T/C and-509 T/C) result in an increase in the expression of the *TGF-β-1* gene, which has been associated with a greater susceptibility to various diseases such as myocardial infarction, coronary heart disease, and others [29].

Interferon-gamma (IFN-γ) is a cytokine secreted by Th1 cells, and is considered as a pro-inflammatory cytokine. Its levels are elevated in many diseases such as rheumatoid arthritis [30,31,32]. Recently, the abnormal expression of INF-γ was reported to be associated with a variety of auto-inflammatory and immune diseases [13,14]. IFN-γ can activate inactive CD4+ cells to differentiate into Th1 cells and inhibit the proliferation of Th2.

This work did not find any association of these SNPs with TA in Mexicans patients. Although comparisons were made with different genetic models (dominant, codominant and recessive), no association was found among them. The same was found when performing the haplotype analysis. However, our frequencies—allelic and genotypic—were not significantly different when compared to the frequencies reported in other studies in the Mexican population [31,32]. However, many reports show discrepancies in the frequencies in certain polymorphisms with respect to our present results. For example, the allele frequency of the three polymorphisms studied for IL-10 was different in our study population in comparison to the frequency in a Korean population (*p* ≤ 0.001) [33]. A similar result was found when comparing our population against a Tunisian population [34]. These results indicate that the genetic charge given by ethnicity can be a predisposing factor to the disease, in addition to other factors such as age, sample size, diet or environmental factors including exposure to certain pathogens such as *Mycobacterium tuberculosis*. All of these factors may be contributing differently to Takayasu’s arteritis. There are studies in which the participation of these mycobacteria in the pathogenesis of the disease has been suggested [35]; nevertheless, there are still discrepancies.

On the other hand, it is known that histocompatibility genes have been associated with genetic susceptibility to develop Takayasu’s arteritis. Previous studies in our laboratory showed that human leukocyte antigens (HLA) −B39, B15 and B40 are frequent alleles in the Mexican Mestizo population. Subtypes are rare and apparently of recent generation in Mexico, probably by recombination events at the intron 2 level. Subtypes of these alleles appear to be different from those reported in other populations, mainly of Asian origin [36]. Further, the analysis of genes B * 5201 and B * 3902 were associated with amino acid residues of serine and glutamic acid, which may be involved in antigen binding in the HLA molecule. These data suggest that despite the heterogeneity of the HLA-B alleles, most share characteristic alleles of the populations of the American continent, which may explain the differences in the susceptibility to the disease depending on ethnic origin.

TA is an uncommon disease and the true incidence and prevalence is probably underestimated globally. That is probably why TA has been included within the “orphan” diseases. Globally, the estimated incidence is over 1.2–2.6 million per year and it is 100 times higher in Asian countries [37]. In Japan, for instance, one of the countries with a higher prevalence of TA, there is an estimated prevalence of 0.01%, [38]. In Mexico, TA is frequent and only the institutional prevalence is known. Multicentric studies are not feasible. However, the findings of this series provide information on the relevance of ethnicity for this disease and allow us to consider new hypotheses about the mechanisms of inflammatory damage. In previous studies, we found an association with the insertion sequences of the tuberculosis genome [39]. Therefore, we believe that this insertion can be a trigger for the immune response. We also know that the pattern of recognition receptors present in the cells participates in the innate immune system, identifying molecular patterns associated with microbial pathogens. Those associated with damage or danger give rise to the immune response. We believe that the present results add to the knowledge on triggering mechanisms for TA. Further studies are needed to allow us to determine the pathways of damage through inflammatory and anti-inflammatory cytokines, which do not appear to be the main source of perpetuity of the inflammatory response in TA.

## 5. Conclusions

In conclusion, our results suggest that polymorphisms in the *IL-10* −*1082 A/G* (rs1800896), *IL-10* −*819 C/T* (rs1800871), *IL-10* −*592 A/C* (rs1800872), *TNF-α* −*238 A/G* (rs361525), *TNF-α* −*308 A/G* (rs1800629), *IFN-γ* −*179 G/T* (rs2069709), *IFN-γ* −*155 G/A* (rs2069710), *TGF-β* −*509 T/C* (rs 1800469) and *TGF-β 29 T/C* (rs 1800470) genes make no genetic contribution to the susceptibility of TA in the Mexican population. Taking into account the role of these cytokines in inflammatory processes may require additional studies. In addition to investigating the pathways of inflammation, such as oxidative stress or the route of arachidonic acid, the consideration of environmental factors such as previous exposure to infectious agents with a larger sample size needs to be further explored.

## Figures and Tables

**Table 1 ijerph-16-04863-t001:** Polymorphisms studied in Takayasu´s patients.

Gene Localization	Chromosomal	Gene Name	Total SNPs Studied	Marker (db SNP ID)	Site Polymorphic
IL-10	1q31	Interleukin-10	3	rs1800896	IL-10 −1082 A/G
				rs1800871	IL-10 −819 C/T
				rs1800872	IL-10 −592 A/C
TNF-α	6p21	Tumor necrosis factor-alpha	2	rs361525	TNF-α −238 A/G
				rs1800629	TNF-α −308 A/G
IFN-γ	12q15	Interferon-gamma	2	rs2069709	IFN-γ −179 G/T
				rs2069710	IFN-γ −155 G/A
TGF-β	19q13.2	Transforming growth factor beta	2	rs1800469	TGF-β −509 T/C
				rs1800470	TGF-β 29 T/C

SNP—single nucleotide polymorphism; IL—interleukin; TNF—tumor necrosis factor; TGF—transforming growth factor, IFN—interferon gamma, db SNP ID—Database of Single Nucleotide Polymorphisms identification, A/G—adenine/Guanine, C/T—Cytosine/Thymine; rs—Reference SNP 2.3. Statistical Analysis

**Table 2 ijerph-16-04863-t002:** Demographic characteristics of the study population.

Demographic Features	Takayasu’s Patients	Healthy Controls	*p*
Total subjects (n)			
Female	50	48	0.021 ^a^
Male	2	12	
Age (years)	28	34.7	0.158 ^b^
Medium	(13–52)	(17–51)	
Hata’s classification (%)			
Type I	9.6	-	
Type II	7.7	-	
Type III	5.7	-	
Type IV	0	-	
Type V	76.9	-	
Activity (%)			
Yes	61.5	-	
No	38.5	-	
Blood pressure (mmHg ± SD)			
Diastolic	82.93 ± 28.5	78.80 ± 9.6	0.068 ^c^
Systolic	140.34 ± 22.3	122.61 ± 17.3	0.004 ^c^

(mmHg) millimeter of mercury ± (SD) standard deviation; ^a^ chi square test; ^b^ Mann Whitney test; ^c^ T Student Test (*t*-test).

**Table 3 ijerph-16-04863-t003:** Genotype and allele frequencies of the TNF-α, TGF-β, INF-γ and IL-10 polymorphisms in TA patients and controls.

Gene	Genotype	Controls		Patients		*p*	OR (95% CI)
		*n*	%	*n*	%		
*TNF-α-238*	GG	59	98.3	49	94.2	0.335	0.271 (0.02–2.74)
	GA	1	1.7	3	5.8	0.339	0.283 (0.02–2.76)
	AA	0	0	0	0	-	-
	Alleles						
	G	119	99.1	101	97.1	0.339	0.282 (0.02–2.76)
	A	1	0.9	3	2.9	-	-
*TNF-α-308*	GG	51	85	45	86.5	0.819	1.000(0.40–2.45)
	GA	9	15	7	13.5	0.946	0.947 (0.55–1.60)
	AA	0	0	0	0	-	-
	Alleles						
	G	111	92.5	97	93.3	0.970	1012 (0.40–3.18)
	A	9	7.5	7	6.7	-	-
*TGF-β-509*	TT	17	28.3	11	21.1	0.511	0.678 (0.28–1.62)
	TC	23	38.8	26	50.0	0.293	1.608 (0.75–3.41)
	CC	20	33.3	15	28.8	0.759	0.810 (0.36–1.81)
	Alleles						
	T	57	47.5	48	46.1	0.946	0.947 (0.55–1.60)
	C	63	52.5	56	53.4	-	-
*TGF-β 29*	TT	17	28.3	13	25.0	0.854	0.843 (0.36–1.95)
	TC	25	41.7	27	51.9	0.370	1.512 (0.71–3.19)
	CC	18	30	12	23.0	0.541	0.700 (0.29–1.63)
	Alleles						
	T	59	49.1	53	50.9	0.893	1.07 (0.63–1.81)
	C	61	50.9	51	49.1	-	-
*IFN-γ-179*	GG	57	95	50	96.2	0.869	1.315 (0.21–8.19)
	GT	3	5	2	3.8	0.869	0.760 (0.12–4.73)
	TT	0	0	0	0	-	-
	Alleles						
	G	117	97.5	102	98.1	0.871	1.30 (0.21–7.98)
	T	3	2.5	2	1.9	-	-
IFN-γ-155	AA	60	100	52	100	-	-
	AG	0	0	0	0	-	-
	GG	0	0	0	0	-	-
	Alleles					-	-
	A	120	100	104	100	-	-
	G	0	0	0	0	-	-
IL-10-592	AA	8	13.3	5	9.6	0.751	0.691 (0.27–2.62)
	AC	30	50	28	53.8	0.828	1.166 (0.55–2.45)
	CC	22	36.7	19	36.5	0.855	0.994 (0.46–1.59)
	Alleles						
	A	46	38.3	38	36.5	0.890	0.926 (0.53–1.59)
	C	74	61.7	66	63.5	-	-
IL-10-819	TT	8	13.3	6	11.5	1.000	0.847 (0.27–2.62)
	TC	27	45	26	50.0	0.734	1.222 (0.58–2.57)
	CC	25	41.7	20	38.5	0.879	0.875 (0.40–1.86)
	Alleles						
	T	43	35.8	38	36.5	0.976	1.031 (0.59–1.78)
	C	77	64.2	66	63.5	-	-
IL-10-1082	AA	32	53.3	23	49.2	0.440	0.693 (0.32–1.46)
	AG	26	43.3	27	51.9	0.472	1.412 (0.66–2.97)
	GG	2	3.3	2	3.8	0.715	1.160 (0.15–8.53)
	Alleles						
	A	90	75	74	71.1	1.619	0.822 (0.45–1.48)
	G	30	25	30	28.9	-	-

Note: OR—Odds risk, CI—Confidence interval.

**Table 4 ijerph-16-04863-t004:** Haplotype distribution.

IL-10	Haplotype Frequency	TNF-α	Haplotype Frequency	TGF-β	Haplotype Frequency
CCA	0.357	GG	0.911	TC	0.467
ATA	0.352	GA	0.071	CT	0.435
CCG	0.259	AG	0.018	CC	0.065
ACA	0.023			TT	0.033

**Table 5 ijerph-16-04863-t005:** Comparison between allele frequencies in different populations versus Takayasu’s arteritis.

	IL-10 −592		pC	OR (CI 95%)	IL-10 −819		pC	OR (CI 95%)	IL-10 −1082		pC	OR (CI 95%)
	A	C	-	-	T	C	-	-	A	G	-	-
Our study	36.5	63.5	-	-	36.5	63.5	-	-	71.1	28.9	-	-
Korean [33]	78.6	21.4	0.0001	0.20 (0.13–0.31)	73.3	26.7	0.0001	0.20 (0.13–0.32)	93.9	6.1	0.0001	0.16 (0.05–0.27)
Tunisian [34,35,36,37,38,39]	20.0	80	0.0003	96.9 (29.7–31.0)	19.2	80.8	0.0002	2.12 (1.53–3.80)	44.7	55.3	0.0001	3.10 (1.94–4.96)
German [40]	28.2	71.8	0.1060	7.30 (2.20–24.1)	28.2	71.8	0.1065	1.46 (0.94–2.25)	54.7	45.3	0.0019	2.08 (1.32–3.20)
	**TNF-α −238**				**TNF-α −308**							
	G	A	pC	OR (CI 95%)	G	A	pC	OR (CI 95%)	-	-	-	-
Our study	97.1	2.9	-	-	93.7	6.7	-	-	-	-	-	-
Indian [41]	91.6	8.3	0.1117	3.06 (0.87–10.6)	93.0	7.0	0.223	2.53 (0.71–9.02)	-	-	-	-
Taiwanese [42]	82.1	17.8	0.0004	7.30 (2.20–24.1)	98.6	1.4	0.612	0.46 (0.09–2.34)	-	-	-	-
Iranian [43]	25.8	74.2	0.0001	96.9 (29.7–31.6)	92.8	7.2	0.213	2.62 (0.72–9.52)	-	-	-	-
	**TGF-β −509**				**TGF-β −29**							
	T	C	pC	OR (CI 95%)	T	C	pC	OR (CI 95%)	-	-	-	-
Our study	46.1	56.4	-	-	50.9	49.1	0.0010	-	-	-	-	-
Iranian [44]	39.8	60.2	0.3471	1.26 (0.80–0.09)	31.0	69.0	0.8081	2.31 (1.42–3.76)	-	-	-	-
Chinese [45]	-	-	-	-	48.9	51.1	0.9125	1.08 (0.69–1.70)	-	-	-	-
Caucasian [46]	-	-	-	-	49.8	50.2	-	1.04 (0.68–1.61)	-	-	-	-
Taiwanese [47]	44.5	55.4	0.9726	0.97 (0.61–1.56)	-	-	-	-	-	-	-	-
Spanish [48]	66.5	33.5	0.0001	0.43 (0.28–0.65)	-	-	-	-	-	-	-	-
	**IFN-γ −179**				**IFN-γ −155**							
	G	T	pC	OR (CI 95%)	A	G	pC		-	-	-	-
Our study	98.1	1.9	-	-	100	0.0	-	-	-	-	-	-
Chinese [49]	-	-	-	-	100	0.0	-	-	-	-	-	-
Sudanese [50]	99.5	0.5	0.5383	0.24 (0.02–1.74)	-	-	-	-	-	-	-	-

pC—value p corrected, OR—Odds risk, CI: Confidence interval.

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
