# Peer review of "Lack of Association between Cytokine Genetic Polymorphisms in Takayasu’s Arteritis in Mexican Patients"

_ijerph, 2019, doi:10.3390/ijerph16234863_

Round 1

Reviewer 1 Report

General comments:

In this paper, Soto et al report that the polymorphism of several cytokines (IL-10, TNF-alpha, TGF-beta and IFN-gamma) is unlikely to be involved in the pathogenesis of Takayasu arteritis in Mexican patients. The findings are of some interest, although the results are negative. However, there are too many errors in English language throughout the text and the tables. Careful proof-reading is required, and the description in the text and the tables should be standardized.

Specific comments:

1) There are too many errors in English language throughout the text and the tables, which should be revised.

For example:

Line 27: “associations not were detected” should be “associations were not detected”.

Line 29: “might not contribution to” should be “might not contribute to”.

Line 36: “can leads to” should be “can lead to”.

Line 60: “G-1082” should be “G-1082A”.

Line 64: “these study” should be “these studies”.

And so on, additional many errors throughout the text and the tables.

2) Table 1: The description should be standardized, and the errors should be revised.

INF-gamma-179 G7T should be INF-gamma-179 G/T.

TGF-beta T29C should be TGF-beta-29 T/C

3) Lines 115-117: The authors mention that 52 patients were examined; 32 cases in active phase and 17 cases in non-active phase. How about the other three cases?

4) Table 3:

IFN-gamma-179: Alleles; A should be T.

The order of cytokines described should be standardized for all tables.

5) HLA-B52 has been shown to be associated with Takayasu arteritis. Did the authors examine into HLA types of these Mexican patients? It is preferable to provide some discussion about HLA types in association with this arteritis.

Author Response

Thank you very much for your observations. 

Reviewer 2 Report

The authors made a study about genetic polymorphisms in some pro-inflammatory and anti-inflammatory cytokines in patients affected by Takayasu arteritis (TA) in Mexico. The authors conclude that there are no differences between patients and control in the frequency of the polymorphism that they analyzed. 

Issues to be addressed

An English speaking writer must revise the paper. The Authors must perform an extensive revision of the Introduction. Indeed, some parts of the Introduction are quite unclear or are incorrect. For example, in line 50, the Authors write ”... cytokines that attract chemotherapy...”, it can be supposed that they would write ”cytokines that act as chemoattractant”. The Authors must wholly rephrase the last sentence of the Introduction. What do they intend as ”genes’ polymorphism susceptibility to Takayasu”? The authors state that the patients were diagnosed and analyzed according to ACR criteria. However, are available newer criteria with higher sensitivity and specificity (e.g. EULAR/PRINTO/PRES criteria). Is it possible to confirm the patients’ diagnosing according to these criteria? The Authors analyzed the polymorphisms of IL-10, TNF-α, TGF-β, and INF-γ. However, during the last two years, also polymorphisms in IL12 and IL17 were associated with TA. The Authors should also analyze these cytokines to complete their study. The description of the patients’ and controls groups does not include any information about possible differences in ethnicity. The Authors should give more information about this issue. At the lines 192-194, the Authors claims the lack of haplotypic differences between patients with active or inactive disease; this is an obvious finding because the disease activity is associated with the moment in which we observe the patient and not to genetic susceptibility. However, it would be essential to analyze the possible differences between patients with different ages at the onset. In table 5, the Authors compare allelic frequencies between their patients and the healthy population of different ethnic groups. However, they had also to compare the differences between their control group and these distinct populations to exclude that the observed discrepancies are due to ethnicities and not linked to disease susceptibility. On line 195 the term ”race” must be substituted by ”ethnicity”. The authors state that this is the first paper about cytokines’ polymorphism and TA, but this is not true, for example, Yang KQ, Yang YK, Meng X, Zhang Y, Jiang XJ, Wu HY, Zhang HM, Song L, Bian J, Wen D, Wang LP, Zhou XL. Lack of association between polymorphisms in interleukin (IL)-12, IL-12R, IL-23, IL-23R genes and Takayasu arteritis in a Chinese population. Inflamm Res. 2016 Jul;65(7):543-50. The Authors must revise their Bibliography because some papers about TA genetic susceptibility were published after 2017.

Author Response

Thank ypou very much for your observations.

Reviewer 3 Report

This manuscript suggests that there is no any significant prevalence of SNPs tested of the cytokines in TAk patients from Mexico. Although the manuscript depicts useful information that no SNPs tested are associated with Takayasu’s Arteritis (TAK), the article lacks information about rationale on carrying out this experiments (Abstract itself starts with AIM, rather than the rationale). TAK involves inflammation in the artery walls, and could be treated using the inhibitors of inflammatory cytokines such as anti-TNF. So how does it make sense to see SNPs in TNF? Authors need to provide rationale on carrying out this research.

There are many grammatical errors in the manuscript which I find impractical to list those all. The article needs to go through English language check before being accepted for publication.

Author Response

Thank you for your comments.  The manuscript has been reviewed and corrected by an English lenguage native.  

Round 2

Reviewer 1 Report

General comments:

The revision of this manuscript is insufficient. There are still too many errors in English language throughout the text and the tables, and the description of the findings is inaccurate, which may make this manuscript unreliable.

I recommend that the manuscript should be checked out by native English speakers, and the findings and the description should be carefully checked again.

The revision may include the following items:

Line 65: “are associated to” should be “are associated with”.

Line 79: “femoral artery were” should be “femoral artery) were”.

Line 80: “is classified by the)” should be deleted.

Line 90: DM2 should be described in full spelling.

Line 129: “67.3%” should be “61.5%”.

Line 130: “32.7%” should be “38.5%”.

Table 2: The subjects of healthy controls: “2” male should be “12” male.

Table 3: Title: “TK” patients should be “TA” patients.

Table 3: IFN-gamma-179: Genotype column: Alleles: “A” should be “T”.

Lines 189-195: The sentences are incomplete, and it is hard to interpret the findings. The findings should be described accurately.

Line 252: RAS should be described in full spelling.

Line 296: “AT” should be “TA”.

Line 391: The journal name is missing.

Finally, it should be noted that there are additional too many errors.

Author Response

Thank you for your comments.  We have carefully reviewed the manuscript and the suggested changes  have been made. (marked in red). in addition, the manuscript has been reviewed by a native of the English lenguage

Reviewer 2 Report

Accept in present form.

Author Response

Thank you for your comments

Reviewer 3 Report

n/a

Author Response

Thank you for your comments

This manuscript is a resubmission of an earlier submission. The following is a list of the peer review reports and author responses from that submission.